# Adversarial Scene Editing:
# Automatic Object Removal from Weak Supervision

**Rakshith Shetty[1]**      **Mario Fritz[2]**      **Bernt Schiele[1]**

[1]Max Planck Institute for Informatics, Saarland Informatics Campus
[2]CISPA Helmholtz Center i.G., Saarland Informatics Campus
Saarbrücken, Germany
[1]`firstname.lastname@mpi-inf.mpg.de`
[2]`firstname.lastname@cispa.saarland`

## Abstract

While great progress has been made recently in automatic image manipulation, it has been limited to object centric images like faces or structured scene datasets. In this work, we take a step towards general scene-level image editing by developing an automatic interaction-free object removal model. Our model learns to find and remove objects from general scene images using image-level labels and unpaired data in a generative adversarial network (GAN) framework. We achieve this with two key contributions: a two-stage editor architecture consisting of a mask generator and image in-painter that co-operate to remove objects, and a novel GAN based prior for the mask generator that allows us to flexibly incorporate knowledge about object shapes. We experimentally show on two datasets that our method effectively removes a wide variety of objects using weak supervision only.

## 1   Introduction

Automatic editing of scene-level images to add/remove objects and manipulate attributes of objects like color/shape etc. is a challenging problem with a wide variety of applications. Such an editor can be used for data augmentation [1], test case generation, automatic content filtering and visual privacy filtering [2]. To be scalable, the image manipulation should be free of human interaction and should learn to perform the editing without needing strong supervision. In this work, we investigate such an automatic interaction free image manipulation approach that involves editing an input image to remove target objects, while leaving the rest of the image intact.

The advent of powerful generative models like generative adversarial networks (GAN) has led to significant progress in various image manipulation tasks. Recent works have demonstrated altering facial attributes like hair color, orientation [3], gender [4] and expressions [5] and changing seasons in scenic photographs [6]. An encouraging aspect of these works is that the image manipulation is learnt without ground truth supervision, but with using unpaired data from different attribute classes. While this progress is remarkable, it has been limited to single object centric images like faces or constrained images like street scenes from a single point of view [7]. In this work we move beyond these object-centric images and towards scene-level image editing on general images. We propose an automatic object removal model that takes an input image and a target class and edits the image to remove the target object class. It learns to perform this task with only image-level labels and without ground truth target images, i.e. using only unpaired images containing different object classes.

Our model learns to remove objects primarily by trying to fool object classifiers in a GAN framework. However, simply training a generator to re-synthesize the input image to fool object classifiers leads to degenerate solutions where the generator uses adversarial patterns to fool the classifiers. We

address this problem with two key contributions. First we propose a two-stage architecture for our generator, consisting of a mask generator, and an image in-painter which cooperate to achieve removal. The mask generator learns to fool the object classifier by masking some pixels, while the in-painter learns to make the masked image look realistic. The second part of our solution is a GAN based framework to impose shape priors on the mask generator to encourage it to produce compact and coherent shapes. The flexible framework allows us to incorporate different shape priors, from randomly sampled rectangles to unpaired segmentation masks from a different dataset. Furthermore, we propose a novel locally supervised real/fake classifier to improve the performance of our in-painter for object removal. Our experiments show that our weakly supervised model achieves on par results with a baseline model using a fully supervised Mask-RCNN [8] segmenter in a removal task on the COCO [9] dataset.

An important use-case of our system would be in automatic content filtering, e.g. for privacy or parental control. This would involve automatic removal of objects and sensitive content from large databases or continuous streams of images. Content to be removed in these scenarios are often personalized and beyond the usually studied object categories in computer vision. Thus a system which can learn to remove these objects from cheap image-level labels would be useful. We demonstrate the applicability of our object remover model to such content filtering task, by training it to automatically remove brand logos from images with only image level labels.

## 2 Related work

**Generative adversarial networks.** Generative adversarial networks (GAN) [10] are a framework where a generator learns by competing in an adversarial game against a discriminator network. The discriminator learns to distinguish between the real data samples and the "fake" generated samples. The generator is optimized to fool the discriminator into classifying generated samples as real. The generator can be conditioned on additional information to learn conditional generative models [11].

**Image manipulation with unpaired data.** A conditional GAN based image-to-image translation system was developed in [12] to manipulate images using paired supervision data. Li et al. [6] alleviated the need for paired supervision using cycle constraints and demonstrated translation between two different domains of unpaired images including (horse↔zebras) and (summer↔winter). Similar cyclic reconstruction constraints were extended to multiple domains to achieve facial attributes manipulation without paired data [5]. Nevertheless these image manipulation works have been limited to object centric images like faces [5] or constrained images like street scenes from one point of view [6]. In our work we take a step towards general scene-level manipulation by addressing the problem of object removal from generic scenes. Prior works on scene-level images like the COCO dataset have focused on synthesizing entire images conditioned on text [13–15] and scene-graphs [16]. However generated image quality on scene-level images [16] is still significantly worse than on structured data like faces [17]. In contrast we focus on the manipulation of parts of images rather than full image synthesis and achieve better image quality and control.

**Object removal.** We propose a two-staged editor with a mask-generator and image in-painter which jointly learn to remove the target object class. Prior works on object removal focus on algorithmic improvements to in-painting while assuming users provide the object mask [18–20]. One could argue that object segmentation masks can be obtained by a stand alone segmenter like Mask-RCNN [8] and just in-paint this masked region to achieve removal. However, this needs expensive mask annotation to supervise the segmentation networks for every category of image entity one wishes to remove for example objects or brand logos.Additionally, as we show in our experiments, even perfect segmentation masks are not sufficient for perfect removal. They tend to trace the object shapes too closely and leave object silhouettes giving away the object class. In contrast, our model learns to perform removal by jointly optimizing the mask generator and the in-painter for the removal task with only weak supervision from image-level labels. This joint optimization allows the two components to cooperate to achieve removal performance on par with a fully supervised segmenter based removal.

## 3 Learning to remove objects

We propose an end-to-end model which learns to find and remove objects automatically from images without any human interaction. It learns to perform this removal with only access to image-level labels without needing expensive ground-truth location information like bounding boxes or masks.

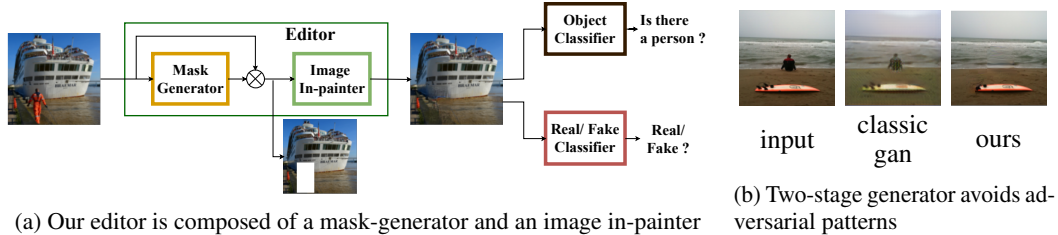

(a) Our editor is composed of a mask-generator and an image in-painter

(b) Two-stage generator avoids adversarial patterns

Figure 1: Illustrating (a) the proposed two-staged architecture and (b) the motivation for this approach

Additionally, we do not have ground-truth target images showing the expected output image with the target object removed since it is infeasible to obtain such data in general.

We overcome the lack of ground-truth location and target image annotations by designing a generative adversarial framework (GAN) to train our model with only unpaired data. Here our editor model learns from weak supervision from three different classifiers. The model learns to locate and remove objects by trying to fool an object classifier. It learns to produce realistic output by trying to fool an adversarial real/fake classifier. Finally, it learns to produce realistic looking object masks by trying to fool a mask shape classifier. Let us examine these components in detail.

## 3.1 Editor architecture: A two-staged approach

Recent works [4, 5] on image manipulation utilize a generator network which takes the input image and synthesizes the output image to reflect the target attributes. While this approach works well for structured images of single faces, we found in own experiments that it does not scale well for removing objects from general scene images. In general scenes with multiple objects, it is difficult for the generator to remove only the desired object while re-synthesizing the rest of the image exactly. Instead, the generator finds the easier solution to fool the object classifier by producing adversarial patterns. This is also facilitated by the fact that the object classifier in crowded scenes has a much harder task than a classifier determining hair-colors in object centric images and thus is more susceptible to adversarial patterns. Figure 1b illustrates this observation, where a single stage generator from [5] trying to remove the person, fools the classifier using adversarial noise. We can also see that the colors of the entire image have changed even when removing a single local object.

We propose a two-staged generator architecture shown in Figure 1a to address this issue. The first stage is a mask generator, $G_M$, which learns to locate the target object class, $c_t$, in the input image $x$ and masks it out by generating a binary mask $m = G_M(x, c_t)$. The second stage is the in-painter, $G_I$, which takes the generated mask and the masked-out image as input and learns to in-paint to produce a realistic output. Given the inverted mask $\widetilde{m} = 1 - m$, final output image $y$ is computed as

$$y = \widetilde{m} \cdot x + m \cdot G_I \left( \widetilde{m} \cdot x \right) \tag{1}$$

The mask generator is trained to fool the object classifier for the target class whereas the in-painter is trained to only fool the real/fake classifier by minimizing the loss functions shown below.

$$L_{\text{cls}}(G_M) = -\mathbb{E}_x \left[ \log(1 - D_{cls}(y, c_t)) \right] \tag{2}$$

$$L_{\text{rf}}(G_I) = -\mathbb{E}_x \left[ D_{\text{rf}}(y) \right] \tag{3}$$

where $D_{cls}(y, c_t)$ is the object classifier score for class $c_t$ and $D_{\text{rf}}$ is the real/fake classifier.

Here $D_{\text{rf}}$ is adversarial, i.e. it is constantly updated to classify generated samples $y$ as "fake". The object classifier $D_{cls}$ however is not adversarial, since it leads to the classifier using the context to predict the object class even when the whole object is removed. Instead, to make the $D_{cls}$ robust to partially removed objects, we train it on images randomly masked with rectangles. The multiplicative configuration in (1) makes it easy for $G_M$ to remove the objects by masking them out. Additionally, the in-painter also does not produce adversarial patterns as it is not optimized to fool the object classifier but only to make the output image realistic. The efficacy of this approach is illustrated in the image on the right on Figure 1b, where our two-staged model is able to cleanly remove the person without affecting the rest of the image.

## 3.2 Mask priors

While the two-stage architecture avoids adversarial patterns and converge to desirable solutions, it is not sufficient. The mask generator can still produce noisy masks or converge to bad solutions like masking most of the image to fool the object classifier. A simple solution is to favor small sized masks. We do this by simply minimizing the exponential function of the mask size, $\exp(\Sigma_{ij} m_{ij})$. But this only penalizes large masks but not noisy or incoherent masks.

To avoid these degenerate solutions, we propose a novel mechanism to regularize the mask generator to produce masks close

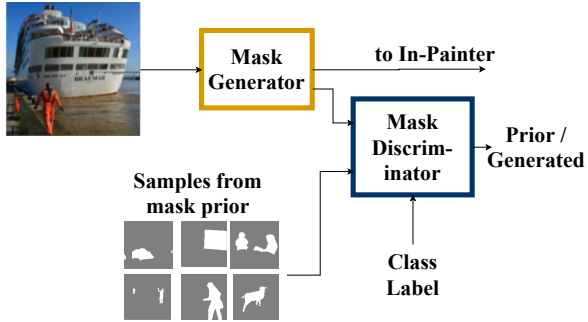

Figure 2: Imposing mask priors with a GAN framework

to a prior distribution. We do this by minimizing the Wasserstein distance between the generated mask distribution and the prior distribution $P(m)$ using Wasserstein GAN (WGAN) [21] as shown in Figure 2. The WGAN framework allows flexibility while choosing the prior since we only need samples from the prior and not a parametric form for the prior.

The prior can be chosen with varying complexity depending on the amount of information available, including knowledge about shapes of different object classes. For example we can use unpaired segmentation masks from a different dataset as a shape prior to the generator. When this is not available, we can impose the prior that objects are usually continuous coherent shapes by using simple geometric shapes like randomly generated rectangles as the prior distribution.

Given a class specific prior mask distribution, $P(m^p|c_t)$, we setup a discriminator, $D_M$ to assign high scores to samples from this prior distribution and the masks generated by $G_M(x, c_t)$. The mask generator is then additionally optimized to fool the discriminator $D_M$. The adversarial losses minimized by $D_M$ and $G_M$ are as below:

$$L(D_M) = \mathbb{E}_x \left[ D_M(G_M(x, c_t), c_t) \right] - \mathbb{E}_{m^p \sim P(m^p|c_t)} \left[ D_M(m^p, c_t) \right] \tag{4}$$

$$L_{\text{prior}}(G_M) = - \mathbb{E}_x \left[ D_M(G_M(x, c_t), c_t) \right] \tag{5}$$

## 3.3 Optimizing the in-painting network for removal

The in-painter network $G_I$ is tasked with synthesizing a plausible image patch to fill the region masked-out by $G_M$, to produce a realistic output image. Similar to prior works on in-painting [22–24], we train $G_I$ with self-supervision by trying to reconstruct random image patches and weak supervision from fooling an adversarial real/fake classifier. The reconstruction loss encourages $G_I$ to keep consistency with the image while the adversarial loss encourages it to produce sharper images.

**Reconstruction losses.** To obtain self-supervision to the in-painter we mask random rectangular patches $m^r$ from the input and ask $G_I$ to reconstruct these patches. We minimize the $L_1$ loss and the perceptual loss [25] between the in-painted image and the input as follows:

$$L_{\text{recon}}(G_I) = \| G_I (\widetilde{m}^r \cdot x) - x \|_1 + \Sigma_k \| \phi_k (G_I (\widetilde{m}^r \cdot x)) - \phi_k(x) \|_1 \tag{6}$$

**Mask buffer.** The masks generated by $G_M(x, c_t)$ can be of arbitrary shape and hence the in-painter should be able to fill in arbitrary holes in the image. We find that the in-painter trained only on random rectangular masks performs poorly on masks generated by $G_M$. However, we cannot simply train the in-painter with reconstruction loss in (6) on masks generated by $G_M$. Unlike random masks $m^r$ which are unlikely to align exactly with an object, generated masks $G_M(x, c_t)$ overlap the objects we intend to remove. Using reconstruction loss here would encourage the in-painter to regenerate this object. We overcome this by storing generated masks from previous batches in a *mask buffer* and randomly applying them on images from the current batch. These are not objects aligned anymore due to random pairing and we train the in-painter $G_I$ with the reconstruction loss, allowing it to adapt to the changing mask distribution produced by the $G_M(x, c_t)$.

**Local real/fake loss.** In recent works on in-painting using adversarial loss [22–24], in-painter is trained adversarially against a classifier $D_{\text{rf}}$ which learns to predict global "real" and "fake" labels for input $x$ and the generated images $y$ respectively. A drawback with this formulation is that only a small

percentage of pixels in the output $y$ is comprised of truly "fake" pixels generated by the in-painter, as seen in Equation (1). This is a hard task for the classifier $D_{rf}$ hard since it has to find the few pixels that contribute to the global "fake" label. We tackle this by providing local pixel-level real/fake labels on the image to $D_{rf}$ instead of a global one. The pixel-level labels are available for free since the inverted mask $\widetilde{m}$ acts as the ground-truth "real" label for $D_{rf}$. Note that this is different from the patch GAN [12] where the classifier producing patch level real/fake predictions is still supervised with a global image-level real/fake label. We use the least-square GAN loss [26] to train the $D_{rf}$, since we found the WGAN loss to be unstable with local real/fake prediction. This is because, $D_{rf}$ can minimize the WGAN loss with assigning very high/low scores to one patch, without bothering with the other parts of the image. However, least-squares GAN loss penalizes both very high and very low predictions, thereby giving equal importance to different image regions.

$$L(D_{rf}) = \frac{1}{\Sigma_{ij}\widetilde{m}_{ij}} \sum_{ij} \widetilde{m}_{ij} \cdot (D_{rf}(y)_{ij} - 1)^2 + \frac{1}{\Sigma_{ij}m_{ij}} \sum_{ij} m_{ij} \cdot (D_{rf}(y)_{ij} + 1)^2 \qquad (7)$$

**Penalizing variations.** We also incorporate the style-loss ($L_{sty}$) proposed in [24] to better match the textures in the in-painting output with that of the input image and the total variation loss ($L_{tv}$) since it helps produce smoother boundaries between the in-painted region and the original image.

The mask generator and the in-painter are optimized in alternate epochs using gradient descent. When the $G_M$ is being optimized, parameters of $G_I$ are held fixed and vice-versa when $G_I$ is optimized. We found that optimizing both the models at every step led to unstable training and many training instances converged to degenerate solutions. Alternate optimization avoids this while still allowing the mask generator and in-painter to co-adapt. The final loss function for $G_M$ and $G_I$ is given as:

$$L_{total}(G_M) = \lambda_c L_{cls} + \lambda_p L_{prior} + \lambda_{sz} \exp(\Sigma_{ij}m_{ij}) \qquad (8)$$
$$L_{total}(G_I) = \lambda_{rf} L_{rf} + \lambda_r L_{recon} + \lambda_{tv} L_{tv} + \lambda_{sty} L_{sty} \qquad (9)$$

## 4 Experimental setup

**Datasets.** Keeping with the goal of performing removal on general scene images, we train and test our model mainly on the COCO dataset [9] since it contains significant diversity within object classes and in the contexts in which they appear. We test our proposed GAN framework to impose priors on the mask generator with two different priors namely rotated boxes and unpaired segmentation masks. We use the segmentation masks from Pascal-VOC 2012 dataset [27] (without the images) as the unpaired mask priors. To facilitate this we restrict our experiments on 20 classes shared between the COCO and Pascal datasets. To demonstrate that our editor model can generalize beyond objects and can learn to remove to different image entities, we test our model on the task of removing logos from natural images. We use the Flickr Logos dataset [28], which has a training set of 810 images containing 27 annotated logo classes and a test set of 270 images containing 5 images per class and 135 random images containing no logos. Further details about data pre-processing and network architectures is presented in the supplementary material.

**Evaluation metrics.** We evaluate our object removal for three aspects: *removal performance* to measure how effective is our model at removing target objects and *image quality assessment* to quantify how much of the original image is edited and finally *human evaluation* to judge removal.

• **Removal performance:** We quantify the removal performance by measuring the performance of an object classifier on the edited images using two metrics. *Removal success rate* measures the percentage of instances where the editor successfully fools the object classifier score below the decision boundary for the target object class.*False removal rate* measures the percentage of cases where the editor removes the wrong objects while trying to remove the target class. This is again measured by monitoring if the object classifier score drops below decision boundary for other classes.

• **Image quality assessment:** To be useful, our editor should remove the target object class while leaving the rest of the image intact.Thus, we quantify the usefulness by measuring similarity between the output and the input image using three metrics namely peak signal-to-noise ratio (pSNR), structural similarity index (ssim) [29] and perceptual loss [30]. The first two are standard metrics used in image in-painting literature, whereas the perceptual loss [30] was recently proposed as a learned metric to compare two images. We use the squeezenet variant of this metric.

• **Human evaluation:** We conduct a study to obtain human judgments of removal performance. We show hundred randomly selected edited images to a human judge and asked if they see the

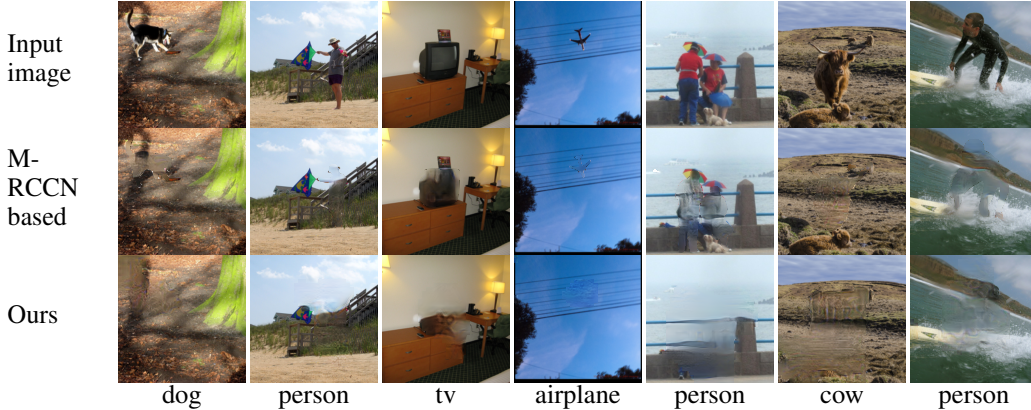

| Input image | | | | | | | |
| M-RCCN based | | | | | | | |
| Ours | | | | | | | |
| dog | person | tv | airplane | person | cow | person |

Figure 3: Qualitative examples of removal of different object classes

target object class. To keep the number of annotations reasonable, we conduct the human evaluation only on the person class (largest class). Each image is shown to three separate judges and removal is considered successful when all three humans agree that they do not see the object class. The participants in the study were not aware of the project and were just asked to determine if they see a 'person' (either full body or clear body parts/ silhouettes) in the images shown. The outputs from different models were all shown in the same session to a human judge in a randomized order to prevent biasing the results against latter models. This human study evaluates the removal system holistically and helps verify that the removal performance measured by a classifier is similar to as perceived by the humans, and thus validating the automatic evaluation protocol.

**Baselines with additional supervision.** Since there is no prior work proposing a fully automatic object removal solution, we compare our model against removal using a stand-alone fully supervised segmentation model, Mask-RCNN [8]. We obtain segmentation mask predictions from Mask-RCNN and use our trained in-painter to achieve removal. Additionally we also compare our model to a weakly supervised segmentation method from [31] (referred to as SDI), which learns to segment objects by using ground truth bounding boxes as supervision. Please note that both the above methods use stronger supervision in terms of object bounding boxes (Mask-RCNN and SDI) and object segmentation (Mask-RCNN) than our proposed method, which uses only image level labels.

## 5   Results

We present qualitative and quantitative evaluations of our editor and comparisons to the Mask-RCNN based removal. Qualitative results show that our editor model works well across diverse scene types and object classes. Quantitative analysis shows that our weakly supervised model performs on par with the fully supervised Mask-RCNN in the removal task, in both automatic and human evaluation.

### 5.1   Qualitative results

Figure 3 shows the results of object removal performed by our model (last row) on the COCO dataset compared to the Mask-RCNN baseline. We see that our model works across diverse scene types, with single objects (columns 1-4) or multiple instances of the same object class (col. 5-6) and even for a fairly large object (last column). Figure 3 also highlight the problems with simply using masks from a segmentation model, Mask-RCNN, for removal. Mask-RCNN is trained to accurately segment the objects and thus the masks it produces very closely trace the object boundary, too closely for removal purposes. We can clearly see the silhouettes of objects in all the edited images on the second row. These results justify our claim that segmentation annotations are not needed to learn to remove objects and might not be the right annotations anyway.

Our model is not tied to notion of objectness and can be easily extended to remove other image entities. The flexible GAN based mask priors allow us to use random rectangular boxes as priors when object shapes are not available. To demonstrate this we apply our model to the task of removing brand logos automatically from images. The model is trained using image level labels and box prior. Qualitative examples in Figure 4 shows that our model works well for this task, despite the fairly

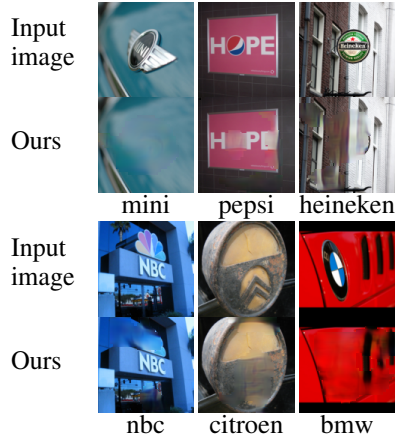

Figure 4: Results of logo removal

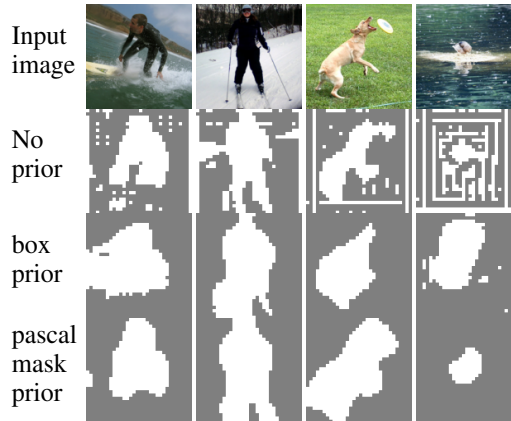

Figure 5: Effect of priors on generated masks

small training set (800 images). It is able to find and remove logos in different contexts with only image level labels. The image on the bottom left shows a failure case where the model fails to realize that the text "NBC" belongs to the logo.

Figure 5 shows the masks generated by our model with different mask priors on the COCO dataset. These examples illustrate the importance of the proposed mask priors. The masks generated by the model using no prior (second row) are very noisy since the model has no information about object shapes and is trying to infer everything from the image level classifier. Adding the box prior already makes the masks much cleaner and more accurate. We can note that the generated masks are "boxier" while not strictly rectangles. Finally using unpaired segmentation masks from the pascal dataset as shape priors makes the generated masks more accurate and the model is able to recover the object shapes better. This particularly helps in object with diverse shapes, for example people and dogs.

## 5.2 Quantitative evaluation of removal performance

To quantify the removal performance we run an object classifier on the edited images and measure its performance. We use a separately trained classifier for this purpose, not the one used in our GAN training, to fairly compare our model and the Mask-RCNN based removal.

**Sanity of object classifier performance.** The classifier we use to evaluate our model achieves per-class average F1-score of 0.57, overall average F1-score of 0.67 and mAP of 0.58. This is close to the results achieved by recent published work on multi-label classification [32] on the COCO dataset, which achieves class average F1-score of 0.60, overall F1-score of 0.68 and mAP of 0.61. While these numbers are not directly comparable (different image resolution, different number of classes), it shows that our object classifier has good performance and can be relied upon. Furthermore, human evaluation shows similar results as our automatic evaluation.

**Effect of priors.** Table 1 compares the different versions of our model using different priors. The box prior uses randomly generated rectangles of different aspect ratios, area and rotations. The *Pascal (n)* prior uses $n$ randomly chosen unpaired segmentation masks for each class from the Pascal dataset. The table shows metrics measuring the removal performance, image quality and mask accuracy. The arrows ↑ and ↓ indicate if higher or lower is better for the corresponding metric. Comparing removal performance in Table 1 we see that while the model with no prior achieves very high removal rate (94%), but it does so with large masks (37 %) which causes low output image quality. As we add priors, the generated masks become smaller and compact. We also see that mIou of the masks increase with stronger priors (0.22-0.23 for pascal prior), indicating they are more accurate. Smaller and more accurate masks also improve the image quality metrics and false removal rates which drop more than half from 36% to 16%. This is inline with the visual examples in Figure 5, where model without prior produces very noisy masks and quality of the masks improve with priors.

Another interesting observation from Table 1 is that using very few segmentation masks from pascal dataset leads to a drop in removal success rate, especially for the *person* class. This is because the

Table 1: Quantifying the effect of using more accurate mask priors

| Prior | Removal Performance | | | Image quality metrics | | | Mask accuracy | |
|---|---|---|---|---|---|---|---|---|
| | removal success ↑ | | false ↓ removal | percep. loss ↓ | pSNR ↑ | ssim ↑ | mIou ↑ | % masked area ↓ |
| | all | person | | | | | | |
| None | **94** | **96** | 36 | 0.13 | 19.97 | 0.743 | 0.15 | 37.7 |
| boxes | 83 | 88 | 23 | 0.11 | 20.41 | 0.777 | 0.18 | 28.1 |
| pascal (10) | 67 | 59 | 17 | **0.07** | **23.81** | **0.833** | **0.23** | **16.7** |
| pascal (100) | 70 | 75 | **16** | **0.07** | 23.02 | 0.821 | 0.22 | 18.1 |
| pascal (all) | 73 | 81 | **16** | 0.08 | 22.64 | 0.803 | 0.22 | 20.2 |

Table 3: Comparison to ground truth masks and Mask-RCNN baselines.

| Model | Supervision | Removal Performance | | | Image quality metrics | | |
|---|---|---|---|---|---|---|---|
| | | removal success ↑ | | false ↓ removal | percep. loss ↓ | pSNR ↑ | ssim ↑ |
| | | all | person | | | | |
| GT masks | - | 66 | 72 | 5 | **0.04** | **27.43** | **0.930** |
| Mask RCNN | Seg. masks & | 68 | 73 | 6 | 0.05 | 25.59 | 0.900 |
| Mask RCNN (dil. 7x7) | bound boxes | 75 | 77 | 10 | 0.07 | 24.13 | 0.882 |
| ours-pascal | image labels & unpaired masks | 73 | **81** | 16 | 0.08 | 22.64 | 0.803 |

*person* class has very diverse shapes due to varying poses and scales. Using only ten masks in the prior fails to capture this diversity and performs poorly (59%). As we increase the number of mask samples in the prior, removal performance jumps significantly to 81% on the person class. Considering these results, we note that the *pascal all* version offers the best trade-off between removal and image quality due to more accurate masks and we will use this model in comparison to benchmarks.

**Benchmarking against GT and Mask-RCNN.** Table 3 compares the performance of our model against baselines using ground-truth (GT) masks and Mask-RCNN segmentation masks for removal. These benchmarks use the same in-painter as *our-pascal* model. We see that our model outperforms the fully supervised Mask-RCNN masks and even the GT masks in terms of removal (66%& 68% vs 73%). While surprising, this is explained by the same phenomenon we saw in qualitative results with Mask-RCNN in Figure 3. The GT and Mask-RCNN masks for segmentation are too close to the object boundaries and thus leave object silhouettes behind when used for removal. When we dilate the masks produced by Mask-RCNN before using for removal, the performance improves overall and is on par with our model (slightly better in all classes and a bit worse in the person class). The drawback of weak supervision is that masks are a bit larger which leads to bit higher false removal rate (16% ours compared to 10% Mask-RCNN dilated) and lower image quality metrics. However this is still a significant result, given that our model is trained without expensive ground truth segmentation annotation for each image, but instead uses only unpaired masks from a smaller dataset.

**Comparison to weakly supervised segmentation.** We compare to the weakly supervised SDI [31] model in Table 2. We use the the output masks generated by SDI to mask the image and use the in-painter trained with our model to fill in the masked region. Simply using the masks from SDI without dilation results in poor removal performance with only 54% success overall and 45% success on the 'person' class. Upon dilation, the performance improves, but is still significantly worse than our model and Mask-RCNN.

Additionally, SDI method starts from boxes generated by a fully supervised RCNN network and generates segmentation with weak supervision, whereas our model uses only image-level labels and hence is more generally applicable.

**Human evaluation.** We verify our automatic evaluation results using a user study to evaluate removal success as described in Section 4. The human judge-

| Model | dil | Rem. Succ. | |
|---|---|---|---|
| | | all | person |
| SDI: supervised | - | 54 | 45 |
| with GT boxes | 7x7 | 64 | 65 |
| Ours | - | 73 | **81** |

Table 2: Comparison to weakly supervised semantic segmentation model, SDI [31]

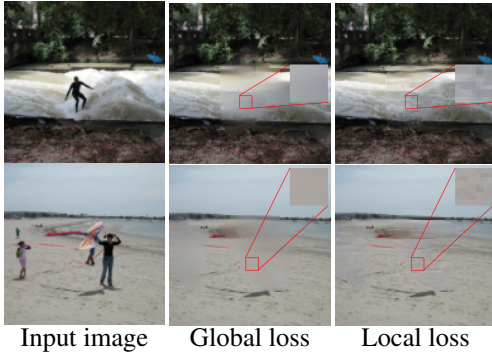

| Input image | Global loss | Local loss |

Figure 6: Comparing global and local GAN loss. Global loss smooth blurry results, while local one produce sharp, texture-rich images.

Table 4: Evaluating in-painting components

| Mask buffer | GAN | TV+ Style | percep. loss ↓ | pSNR ↑ | ssim ↑ |
|---|---|---|---|---|---|
| - | G | - | 0.13 | 20.0 | 0.730 |
| ✓ | G | - | 0.12 | **21.9** | **0.772** |
| ✓ | L | - | **0.10** | 21.5 | 0.758 |
| ✓ | L | ✓ | **0.10** | 21.6 | 0.763 |

Table 5: Joint training helps improve both mask generation and in-painting

| Joint training | Removal success ↑ | mIou ↑ | percep. loss ↓ |
|---|---|---|---|
| - | 0.68 | 0.19 | 0.10 |
| ✓ | **0.73** | **0.22** | **0.08** |

ments of removal performance follow the same trend seen in automatic evaluation, except that human judges penalize the silhouettes more severely.Our model clearly outperforms the baseline Mask-RCNN model without dilation by achieving 68% removal rate compared to only 30% achieved by Mask-RCNN. With dilated masks, Mask-RCNN performs similar to our model in terms of removal achieving 73% success rate.

## 5.3 Ablation studies

**Joint optimization.** We conduct an experiment to test if jointly training the mask generator and the in-painter helps. We pre-train the in-painter using only random boxes and hold it fixed while training the mask generator. The results are shown in Table 5. Not surprisingly, the in-painting quality suffers with higher perceptual loss (0.10 vs 0.08) since it has not adapted to the masks being generated. More interestingly, the mask generator also degrades with a fixed in-painter, as seen by lower mIou (0.19 vs 0.22) and lower removal success rate (0.68 vs 0.73). This result shows that it is important to train both the models jointly to allow them to adapt to each other for best performance.

**In-painting components.** Table 4 shows the ablation of the in-painter network components. We note that the proposed *mask-buffer*, which uses masks from previous batch to train the in-painter with reconstruction loss, significantly improves the results significantly in all three metrics. Using local loss improves the results in-terms of perceptual loss (0.10 vs 0.12) while being slightly worse in the other two metrics. However on examining the results visually in Figure 6, we see that the version with the global GAN loss produces smooth and blurry in-painting, whereas the version with local GAN loss produces sharper results with richer texture. While these blurry results do better in pixel-wise metrics like pSNR and ssim, they are easily seen by the human eye and are not suitable for removal. Finally addition of total variation and style loss helps slighlty improve the pSNR and ssim metrics.

## 6 Conclusions

We presented an automatic object removal model which learns to find and remove objects from general scene images. Our model learns to perform this task with only image level labels and unpaired data. Our two-stage editor model with a mask-generator and an in-painter network avoids degenerate solutions by complementing each other. We also developed a GAN based framework to impose different priors to the mask generator, which encourages it to generate clean compact masks to remove objects. Results show that our model achieves similar performance as a fully-supervised segmenter based removal, demonstrating the feasibility of weakly supervised solutions for the general scene-level editing task.

## Acknowledgments

This research was supported in part by the German Research Foundation (DFG CRC 1223).

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
