[Supplementary Material]

# Adversarial Scene Editing:
# Automatic Object Removal from Weak Supervision

**Rakshith Shetty[1]**      **Mario Fritz[2]**      **Bernt Schiele[1]**

[1]Max Planck Institute for Informatics, Saarland Informatics Campus
[2]CISPA Helmholtz Center i.G., Saarland Informatics Campus
Saarbrücken, Germany
[1]`firstname.lastname@mpi-inf.mpg.de`
[2]`firstname.lastname@cispa.saarland`

## 1   Data pre-processing

We pre-process the COCO dataset to filter out images containing large objects. This is done by removing images with single object class covering more than 30% of the image. Removing very large objects requires the model to hallucinate most of the image, and the task becomes very difficult. This is also reasonable from application point of view, since the object larger than 30% of the image are very often the focus of the image, and removing them would not be very useful. After size filtering we are have 39238 training, 2350 validation and 1905 test images.

In both the datasets, the images are preprocessed by resizing the shortest edge to 128 pixels and center-cropping to obtain 128x128 images. Further data augmentation is performed using random horizontal flips and the images are normalized to have zero mean and unit variance.

Similar preprocessing is applied to the masks from the Pascal dataset to obtain 128x128 dimensional masks for the prior. The Pascal dataset has about 215 masks on average for each of the 20 classes. This gives us 4318 masks for our mask-prior, which is much lower compared to 39238 training images in our pre-processed dataset.

## 2   Network architectures

**Mask generator.** The mask generator architecture is built on top of a VGG network pre-trained on Image net classification task. Note that this pre-training also uses only image-level labels. We use the features from the convolutional layer before the first fully connected layer from the VGG-19 architecture as our starting point. Additionally, we remove two previous maxpool layers to obtain features of dimensions $512 \times 32 \times 32$. This is concatenated with 20 dimensional one-hot vector representing the target class. On top of this we add the following layers:

$$C_{512}^3 - L_{0.1} - R_{512} - C_{256}^3 - L_{0.1} - R_{256} - C_{128}^3 - L_{0.1} - R_{128} - C_{21}^7 - S$$

where $C_n^k$ indicates a convolutional layer with $n$ filters of $k \times k$ size and $R_n$ is a residual block with $n$ filters, $L_{0.1}$ is leaky relu non-linear activation layer and S is the sigmoid activation function.

Each residual block is $R_n$ consists of

$$C_n^3 - I_n - L_{0.1} - C_n^3 - I_n - L_{0.1}$$

where $I_n$ is a $n$ dimensional instance normalization layer. with slope parameter 0.1. We also concatenate the target class vector after every residual block since it improves the performance.

**In-painter.** Our in-painter network architecture is designed borrowing ideas from prior works [1, 2]. We use the same basic architectures as in [1] but incorporate dilated convolutions in the bottleneck

Table 1: Comparing our inpainter to state-of-the art methods. Other results are taken from the paper [4].

| Metric | Method | Relative mask size | | | | | |
|--------|--------|------------|------------|------------|------------|------------|------------|
| | | (0-0.1) | [0.1-0.2) | [0.2-0.3) | [0.3-0.4) | [0.4-0.5) | [0.5-0.6) |
| SSIM | PM [5] | 0.947 | 0.865 | 0.768 | 0.675 | 0.579 | 0.472 |
| | GL [2] | 0.923 | 0.829 | 0.721 | 0.627 | 0.533 | 0.440 |
| | PConv [4] | 0.945 | 0.870 | 0.779 | 0.689 | 0.595 | 0.484 |
| | Ours | **0.972**±0.00 | **0.925**±0.00 | **0.870**±0.00 | **0.813**±0.00 | **0.758**±0.00 | **0.721**±0.01 |
| pSNR | PM [5] | 33.68 | 27.51 | 24.35 | 22.05 | 20.58 | 18.22 |
| | GL [2] | 29.74 | 23.83 | 20.73 | 18.61 | 17.38 | 16.37 |
| | PConv [4] | **34.34** | **28.32** | **25.25** | **22.89** | **21.38** | **19.04** |
| | Ours | 32.68±0.09 | 26.07±0.03 | 22.84±0.03 | 20.67±0.02 | 19.09±0.04 | 18.14± 0.17 |

layers to improve performance on larger masks as proposed in [2]. The in-painter takes the input image concatenated with the mask and is agnostic to the object class.

The in-painter network is built with a *downsampling* block, *bottle-neck* block and the *upsampling* block. The layers in the *downsampling* block are

$$C_{64}^4 - I_{64} - L_{0.1} - D_{128}^2 - I_{128} - L_{0.1} - D_{256}^2 - I_{256} - L_{0.1} - D_{512}^2 - I_{512} - L_{0.1}$$

where $D_n^2$ is a downsampling layer halving the spatial dimensions of the input. It consists of a convolutional layer with filter size $4 \times 4$ and stride 2. The *bottle-neck* block is just six back-to-back residual layers, $R_{256}$. Finally, the *upsampling* block is made of three upsampling blocks followed by an output convolutional layer with tanh non-linearity ($T$)

$$U^2 - C_{256}^3 - I_{256} - L_{0.1} - U^2 - C_{128}^3 - I_{128} - L_{0.1} - U^2 - C_{64}^3 - I_{64} - L_{0.1} - C_3^7 - T$$

where $U_2$ is a bilinear up-sampler which doubles the spatial dimensions of the input.

**Object Classifier.** Our object classifier is designed on top off the VGG-19 network backbone which is pre-trained on Imagenet. We add the following layers after the last convolutional layer of VGG-19

$$C_{512}^3 - B_{512} - L_{0.1} - C_{512}^3 - B_{512} - L_{0.1} - G - \text{Lin}_{20} - S$$

where $B_n$ is a n-dimensional batch normalization layer, $G$ is a global pooling layer and $Lin_{20}$ is a linear layer mapping input to 20 class scores.

# 3 Implementation Details

All the components of our model are trained using stochastic gradient descent with Adam optimizer. The adversarial discriminators used in the mask prior and real/fake classification are regularized with the gradient penalty [3]. The model hyper-parameters were chosen using a validation set. The final settings of the hyper-parameters in our loss function were $\lambda_c = 12$, $\lambda_p = 3$, $\lambda_{sz} = 18$, $\lambda_{rf} = 2$, $\lambda_r = 100$, $\lambda_{tv} = 10$, and $\lambda_{sty} = 3000$.

We implement all our models with the PyTorch framework. We will release our code upon publication.

# 4 Comparing In-painter to state-of-the-art

We compare our in-painter model to state-of-the-art image in-painting models from literature on the Places2 [6] dataset in Table 1. Results from the other papers shown in Table 1 are taken from [4]. Since we do not have access to the masks used to test these models, we test ours using random rotated rectangular masks. We mask the input image with up to four randomly generated rectangular masks and measure the in-painting performance. From the table we see that our model performs better than prior work in-terms of structural similarity index (ssim) and is bit worse than partial convolution based method [4] (PConv) in terms of pSNR.

We have noticed in our experiments that irregular masks are much harder to in-paint than axis-aligned rectangular masks. Our model learns to handle both since it is trained with the mask-generator.

Figure 1: Qualitative comparison of our in-painter to GL [2]. We compare the performance of the two models with axis aligned and rotated rectangular patches. These results show that the inpainters trained with only regular axis-aligned rectangles (as in GL) do not perform well with even a small change to the mask distribution including on rotated rectangular masks.

However, prior work [2] is only trained with axis-aligned rectangles (it does not allow easy extension to irregular masks as the local discriminator needs a bounding box) and performs poorly on irregular masks. We demonstrate this in qualitative examples shown in Figure 1. We can see from these examples that while [2] works well on axis-aligned rectangles, it degrades with rotated rectangles. This is the primary reason our model has an advantage over [2] in the metrics shown in Table 1 and is on par with [4] (better in SSIM worse in pSNR), which explicitly handles irregular masks.

## 5 Qualitative Results

Figure 2 shows more qualitative examples of objects removal on the COCO dataset. Examples in the first three rows showcase a wide-variety of scenes where our model is able to successfully remove the target object class. It works for objects of different poses and sizes, and with single or multiple instances of the target object class.

Last row of Figure 2 highlights some failure modes we observed in our model. The First four columns in the last row shows failures where the full extent of the object is not removed and some parts of the object is still visible. However, only in case of the *motorcycle* we can clearly identify the removed object class in the edited image. In case of the *boat* image, only one instance of *boat* has

been removed by the editor. Similarly in the second last column, although the larger *person* has been removed, the smaller instances of people are visible in the background. Finally, the last column shows a case of *false removal* wherein in an attempt to remove the *horse*, our editor also removes the people riding the horse.

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

Figure 2: Qualitative examples of removal of different object classes in diverese scenes. First three rows show examples where our model does well, whereas the last row highlights some failure cases.