[Reviews · NeurIPS 2018]

Reviewer 1



The details provided on the experiment with SDI[2] are somewhat lacking, and the question on how that works combined with standard inpainting was not really addressed. However, they raise a valid point in regards to it requiring box-level supervision compared to the image-level supervision they use. Overall I'm still leaning to accept this submission. ====================================== The paper proposes a system that can learn to remove objects of specific categories from images automatically (e.g. persons). The system is composed of two stages: 1) a mask generator which zero-es out regions in the image such that an image classifier does not recognize the specified category (e.g. it is not classified as person) 2) an inpainting network which infills the masked image so that it looks realistic again Notably, the mask network uses weak supervision: it only can see real masks from a smaller dataset through a GAN loss. More specifically, it is trained to produce masks that hide the target object (fooling an image level classifier) and to also produce masks that are realistic from the view of a mask discriminator. Overall the paper is well written and provides extensive experiments, building a strong baseline for the task from Mask-RCNN + inpainting network (which uses stronger supervision since mask-rcnn is trained on a large corpus of segmentation masks). Qualitatively the network works fairly well, and quantitatively it is competitive with the mask-rcnn baseline. The benefit of the weak supervision is demonstrated by showing that it can work on a dataset that has no segmentation masks available (a logo dataset). The paper makes the point that it is actually helpful to have coarse masks that hide the (to be removed) object's boundaries, compared to fine grained masks. This is demonstrated both when comparing the proposed approach (which has coarse masks because of weak supervision) to the mask-rcnn baseline, and also by improving the mask-rcnn baseline by dilating its segmentation masks. However, based on this I question the need for the the GAN based mask generator. One could interpret the results as that for object removal, you don't really need a very good instance segmentation network (since dilating mask-rcnn helps). Therefore, we should also expect to see good results when using a weakly supervised instance segmentation network. In principle, we can view the first stage of the system as exactly that, but shouldn't the baseline for weak supervision then be as follows: * Standard weakly supervised instance segmentation network, such as (Simple Does It: Weakly Supervised Instance and Semantic Segmentation, Khoreva et al, CVPR 2017 ). * Possibly dilate its masks * Use standard inpainting approach on top So basically, I question the need for the first stage: is it not just a weakly supervised instance segmentation approach? If so, is competitive with established such approaches, and should they not be the baseline? Another concern is what will the whole system be used for? I understand that removing people from images in an unsupervised manner is a difficult problem, but it would be nice to see a concrete example where it is useful. Nontheless, I found the paper overall interesting in terms of the problem studied, the approach and the quality of the experiments, so I'm currently leaning towards accepting it.

Reviewer 2



I have read the other reviews and the author rebuttal and am still in favor of accepting the paper. The rebuttal is impressively thorough, addressing all of my concerns rather decisively with new experimental evidence. ------------------ This paper proposes a new problem, interaction-free object removal, and a neural architecture for solving it. In the problem setup, the system is given an image and an object class label and must remove all instances of that class from the image. The system must thus learn to both localize the object, remove it, and in-paint the removed region. The paper proposes to accomplish this by training a generator network to fool an object classifier into thinking the object class is no longer present in the image. To prevent the generator from doing this by synthesizing adversarial noise patterns, it is divided into two stages: (1) a mask generator and (2) an in-painter. Critically, only the mask generator is trained to fool the classifier; the in-painter is trained only to fool a real/fake image classifier, preventing it from learning to create adversarial noise patterns. The system also regularizes the shape of the generated masks by sampling them from a prior (in the form of a GAN) learned from a separate, unpaired dataset of mask shapes. The system is evaluated by via classification performance and perceptual quality on removing objects from COCO images, and also on removing logos from Flickr Logos images. This paper presents an interesting research direction. I was at first unconvinced about the utility of the 'unsupervised object removal' problem setup, but the logo removal use case turned me around--I can definitely see how there can be scenarios where one wants to remove something from large collections of images (or streams of images) en masse, without any human intervention. The two-stage generator scheme the authors use to avoid adversarial noise patterns is quite clever. I do find the qualitative results of the method to be a bit underwhelming, especially when compared with recently published work on 'supervised' inpainting (and I have some questions about the reported favorable comparison to those methods in the supplement; see below). That said, this is the first work on a new problem, so that could be considered acceptable. Overall, I'm leaning towards wanting to accept this paper. - - - - - - - - - I'm glad to see that supplemental includes comparison against current state-of-the-art inpainting methods; this was something that I felt was missing from the main paper. I would like to see qualitative results for this comparison, as well. Qualitatively, the results from the proposed method just don't look as good as results presented in recent inpainting work--yet the proposed method numerically performs best according to Table 1 in the supplement. I'm having a hard time reconciling this discrepancy. Are the quantitative metrics not very reflective of perceptual quality? Were the prior methods insufficiently trained? Do the authors have any insights, here?

Reviewer 3



Summary ======= This paper describes an approach to automatically removing a class of objects from images. The approach consists of 2 networks, 1 trained to produce a mask, and 1 network trained to inpaint the masked regions. The mask network is trained to fool a classifier detecting the presence of certain objects in the image, and simultaneously to produce masks whose shape distribution matches that of a segmentation dataset (WGAN). The inpainting network is trained adversarially to fool a classifier detecting modified pixels (LS-GAN). Good ==== – Combining a masking and an inpainting network to overcome noisy images is a neat idea – The empirical section is extensive, including ablation studies and human evaluations of masks – The approach seems to work well Bad === – A more elegant solution might have been to also train the mask network to fool the real-fake network (LS-GAN). This would have naturally encouraged small masks and avoided the need to explicitly penalize large masks. It might have reduced the need for a shape prior as well. – Please provide more details on the human evaluation. How many human subjects? Were these naive observers or authors of the paper? – PSNR/SSIM aren't great metrics for evaluating inpainting results since even very good solutions can change drastically at the pixel level. An evaluation with human observers to evaluate inpainting results may have been a better way to spend that resource than to evaluate the mask model's removal performance.